**Data Availability Statement:** All relevant data are within the paper.

**Funding:** This study was supported by National Key R&D Program of China (2017YFD0201900), Key R&D Program of Xinjiang Uygur Autonomous Region (2016B01001-6), the National Natural

# Field evaluation of sunflower as a potential trap crop of *Lygus pratensis* in cotton fields

**Renfu Zhang**[1,2], **Wei Wang**[2], **Haiyang Liu**[2], **Dengyuan Wang**[1]\*, **Ju Yao**[2]\*

**1** College of Agronomy, Key Laboratory of the Pest Monitoring and Safety Control of Crops and Forests of Xinjiang Uygur Autonomous Region, Xinjiang Agricultural University, Urumqi, Xinjiang, PR China, **2** Institute of Plant Protection, Xinjiang Academy of Agricultural Sciences, Key Laboratory of Intergraded Management of Harmful Crop Vermin of China North-western Oasis, Ministry of Agriculture, Urumqi, Xinjiang, PR China

\* 1827841032@qq.com (DW); yaoju500@sohu.com (JY)

## Abstract

The mirid bug *Lygus pratensis* is an important pest of cotton, and is primarily managed through insecticide application. In this study, conducted in Xinjiang (China), we assessed the relative attractiveness of sunflower (*Helianthus annuus*) to *L. pratensis* adults in local cotton plots from 2015–2016 and quantified the associated contribution of inter-planted sunflower strips to suppress field-level *L. pratensis* populations from 2016–2017. Field-plot trials showed that among six combinations of two sunflower varieties (XKZ6 and SH363) and three planting dates (early-, middle- and late-planted), adult abundance of *L. pratensis* was highest on early-planted XKZ6 and attained 3.7–5.8 times higher levels than in neighboring cotton plots. In commercial cotton fields, the combined deployment of sunflower strips at field edges and the periodic application of insecticides directed to those strips was found to (1) reduce the mean abundance of *L. pratensis* population on cotton by 41.9–44.0%, (2) lower the rate of cotton leaf damage by 27.3–30.6% and boll damage by 44.8–46.0%, and (3) increase the number of mature bolls by 7.5%-8.0%. Our work emphasizes how sunflower can be an effective trap crop for *L. pratensis* and that the establishment of sunflower strips could contribute to its effective and environmentally-sound management in cotton crops.

## Introduction

*Lygus pratensis* (L.) (Hemiptera: Miridae) is a common pest on cotton (*Gossypium* spp.), alfalfa (*Medicago sativa*), Chinese date (*Ziziphus jujuba*), grape (*Vitis vinifera*), pear (*Pyrus* spp.), apple (*Malus pumila*) and many other crops. Aside from its broad host range, this mirid bug has a wide geographic distribution and is found throughout Europe, north Africa, the Middle East, northern India, China, and Siberia [1,2]. In China, *L. pratensis* is found mainly in the Xinjiang Uyghur Autonomous Region, and was locally regarded as a major pest on cotton in the 1950s and 1960s [3–5]. Both nymphs and adults of *L. pratensis* feed on vegetative and reproductive parts of cotton plants, thus causing leaf damage, plant stunting and abscission of squares and bolls [6–9]. Occasionally, *L. pratensis* outbreaks caused 100% damage to cotton plants and resulted in significant loss of crop quality and yield [10]. Over the past decades, insecticide-based approaches have suppressed *L. pratensis* population levels and safeguarded

Science Foundation of Xinjiang Uygur Autonomous Region (2019D01A65), the Basic Research Operating Expenses Program of Public Welfare Research Institute from Xinjiang Uygur Autonomous Region (KY2017067).The funders had no role in study design, data collection and analysis, decision to publish, or preparation of the manuscript.

**Competing interests:** The authors have declared that no competing interests exist.

cotton yields [9]. Yet, *L. pratensis* population levels in Xinjiang have increased in recent years after wide-scale adoption of Bt (*Bacillus thuringiensis*) cotton, and this mirid bug has now re-emerged as a major pest of cotton and multiple other crops (e.g., stone fruits) in cotton agro-landscapes [1,11–14]. As chemically-synthesized insecticides continue to be the cornerstone of *L. pratensis* mitigation programs [1,14], non-chemical technologies need to be developed, validated and implemented [15].

As a polyphagous pest, *L. pratensis* disperses from one host plant to another under field conditions, and exhibits variable feeding responses and a marked preference for plants at flowering stage [16,17]. This type of foraging behavior lends itself to devise trap cropping systems; a method of behavioral manipulation based on the pest's host plant preferences [18,19]. By establishing the pest's preferred host plant (i.e., trap plant) close to the main crop, the target pest is diverted from the main crop and pest damage is lowered. Trap cropping been successfully used against multiple economically-important pests, and been successfully applied for the management of many important pests, including several species of mirid bugs (Hemiptera: Miridae) [20]. For instance, the cotton-alfalfa trap crop system relies on alfalfa strips sown within cotton fields to control *Lygus hesperus* Knight in the United States [21–23]. A similar intercropping pattern has been used in Australia to lower infestation levels of *Creontiades dilutus* (Stål) in Australia [24]. Alfalfa has also been used as a trap crop for several species of *Lygus* (e.g., *L. hesperus* and *L. rugulipennis* Poppius) in strawberry (*Fragaria ananassa*) fields in the United States and in Italy [25–27]. In the UK, German chamomile, *Matricaria recutita*, acts as a trap crop for *L. rugulipennis* on strawberry [28] Furthermore, other plant species, such as red clover (*Trifolium pretense*), mugwort (*Artemisia vulgaris*) and sunflower (*Helianthus annuus*), can be used for trapping species of *Lygus* spp. in lettuce (*Lactuca sativa*) and cucumber (*Cucumis sativus*) [29,30]. In Chinese cotton fields, mungbean (*Vigna radiata*) and cowpea (*Vigna unguiculata*) are used as trap crops for *Apolygus lucorum* (Meyer-Dür) and *Adelphocoris suturalis* (Jakovlev), respectively [31,32]. Overall, trap cropping has proven to be an effective means of mirid bug management in various agro-ecosystems worldwide and thus carries ample promise to reduce the current reliance upon chemically-synthesized insecticides to control *L. pratensis*.

In our field survey, *L. pratensis* was found to attain high population levels on sunflower, hinting its potential value as a trap crop for this mirid bug. In this study, we 1) compared *L. pratensis* infestation levels in field plots established with cotton and different varieties and planting dates of sunflower, 2) assessed whether *L. pratensis*' infestation pressure and damage was lowered in cotton fields inter-planted with sunflower strips.

## Materials and methods

### Plants and field sites

In our trials, we used two sunflower varieties: (1) XKZ6, bred by Institute of Crop Research, Xinjiang Academy of Agricultural Reclamation Sciences (Xinjiang, China) and (2) SH363 bred by Gansu Derui Agricultural Science and Technology Co., Ltd. (Gansu Province, China). The cotton variety used was ZM49, bred by the Institute of Cotton Research, Chinese Academy of Agricultural Science (Henan Province, China). All field experiments were conducted in Guzai village (E77º26′92″, N38º55′14″), Tagaerqi township, Shache County in Xinjiang, China.

### *Lygus pratensis* occurrence in cotton and sunflower plots

In 2015, cotton was planted on April 1, and sunflower was sown at three different times: April 1 (early-planted), April 10 (middle-planted), and April 20 (late-planted). In 2016, cotton was sown on April 3, and sunflower was sown on April 3 (early-planted), April 13 (middle-

planted), and April 23 (late-planted). During each year, the following treatments were established: two sunflower varieties (i.e., XKZ6, SH363) each with three planting dates (6 treatments), and cotton with a single planting date (control treatment).

Each of the above treatments was established in 3 field plots (i.e., replicates), totaling 21 plots for the entire experiment. Each plot was 100 $m_2$ (i.e., 10 m wide by 10 m long), and all plots were randomly arranged with 2 m wide vegetation-free aisles between neighboring plots. The seeding density was $2.25 \times 10^5$ seeds per ha for cotton, and $6.5 \times 10^4$ seeds per ha for sunflower. All plots were under flood irrigation with an identical fertilization and watering regime, and no chemical pesticides were applied during the entire growing season. A field survey of *L. pratensis* population was conducted every five days from early June to middle August during both years. In each plot, using five-point sampling method and 20 consecutive plants were checked each point (Fig 1A), the number of *L. pratensis* adults was visually determined and recorded.

Beginning in early June, as these first generation of *L pratensis* nymphs become adults on spring host plants, they disperse to other host plants and then produce second generation there. In other words, before they attack the sunflower and cottons, no *L. pratensis*' infestation were occurred on these two crops. Therefore, we only investigate the relative attractiveness of sunflowers and cottons to adults from entering period, ignore the nymphs.

### *Lygus pratensis* occurrence in cotton with and without sunflower border strips

Cotton and sunflower were simultaneously planted on April 5, 2016 and April 9, 2017. Seeding densities of cotton and sunflower were $2.25 \times 10^5$ seeds per ha and $6.5 \times 10^4$ seeds per ha, respectively. During each year, a total of eight cotton fields (each approximately 30 m wide and 30 m long) were set up with >10 m wide vegetation-free areas between neighboring fields. Four cotton fields with sunflower strips were established as trap cropping treatments, and the other four without sunflower strips were designated as control treatment.

For each trap-cropping field, two 1-m wide strips of sunflower (accounting for 6% crop surface) were planted on the opposite borders of the field, parallel to the cotton rows. Cotton fields with and without sunflower strips were managed under identical irrigation, fertilization

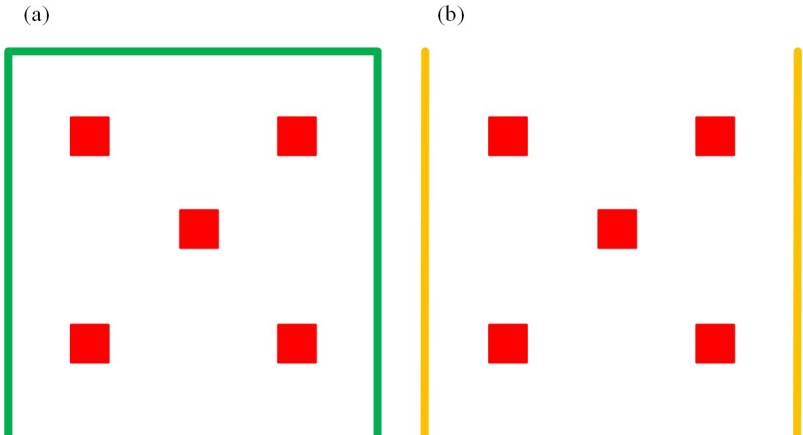

**Fig 1. The population investigate method of *Lygus pratensis* in field.** (a) The left picture showed the investigation method of sunflower with different varieties and planting dates, and cotton in the plots. Within green square represent plot, and red indicates sampling points. b) The right picture showed the investigation method in cotton field with and without sunflower strips. Yellow lines are sunflower strips, and red indicates sampling points.

and crop management schemes. No pesticides were used on cotton plants in any of the fields. To control *L. pratensis* populations on sunflower strips, 40% acetamiprid water dispersible granules at 120g/ha or 25% imidacloprid wettable powder at 120g/ha with water 600 kg/ha were applied in alternation every 10 days following the first detection of the pest.

Sampling of *L. pratensis* in cotton fields was done every five days, from early June to mid-August, using the same method as above. On each sampling date, the number of *L. pratensis* adults and nymphs (per plant) was recorded, and *L. pratensis* feeding damage on the top five cotton leaves was assessed in late June. Also, the number of mature cotton bolls (diameter >2.5 cm) was determined and the damage rate for bolls was determined in late August. For both boll number and damage level, 100 plants from five selected points (20 plants per point) were examined in each field (Fig 1B).

## Statistical analysis

Differences in the abundance of *L. pratensis* (including adults, nymphs, and total population, as three counts) in cotton and sunflower plots (2015–2016) or in cotton fields with/without sunflower strips (2016–2017) were analyzed by a repeated-measures analysis of variance (ANOVA) using SAS PROC MIXED, and the means were compared by the least significant difference test (LSD). The data were $\log_{10}(n+1)$-transformed before analysis. The mean abundances of *L. pratensis* adults in plots of cotton and sunflower during 2015–2016 were compared using one-way ANOVA followed by Tukey's HSD for multiple comparisons. Mean abundances of *L. pratensis* population, the damage rate of leaves and bolls (arcsin-transformed), and the number of mature bolls in cotton fields with/without sunflower strips during 2016–2017 were compared through non-paired t-tests. In 2016 and 2017, *L. pratensis* abundances in both cotton fields were below the existing economic thresholds for this pest (i.e., 20 bugs per 100 cotton plants at blossoming stage; Wang et al. [14]) after middle July. Hence, only population abundance records from early June to mid-July were included in the repeated-measures ANOVA and non-paired t-tests.

## Results

### *Lygus pratensis* occurrence in cotton and sunflower plots

In 2015 and 2016, *L. pratensis* adults attained peak abundance between early June and late July in cotton fields. In June, adult abundance was high on early-planted sunflower while in July, abundance of *L. pratensis* adults was relatively higher on early- and middle-planted sunflower. In plots of late-planted sunflower, adult abundance remained low throughout the whole period (Figs 2 and 3).

Population levels of *L. pratensis* adults differed significantly among all seven treatments in 2015 (repeated measures ANOVA; $F_{(6,12)} = 70.88$, $P < 0.0001$) and 2016 ($F_{(6,12)} = 43.59$, $P < 0.0001$) (Fig 2). Also, mean abundance of *L. pratensis* adults on early-planted XKZ6 was significantly higher than on the other six treatments during both years (one-way ANOVA; 2015: $F_{(6,12)} = 66.37$, $P < 0.0001$; and 2016: $F_{(6,12)} = 107.58$, $P < 0.0001$). More specifically, adult abundance on early-planted XKZ6 was 3.73 times and 5.81 times higher than that on cotton in 2015 and 2016, respectively (Fig 3). Cotton as control treatment, early-plant sunflowers XKZ6 have the highest attraction for *L. pratensis*, so it have more potential as trap crop in cotton fields. Therefore, comparing the abundance of *L. pratensis* between sunflowers and cotton is requisite and meaningful.

### *Lygus pratensis* occurrence in cotton with and without sunflower strips

As results above showed, early-plant sunflowers have more attractiveness to *L. pratensis* adults, so used early-plant sunflower strips in cotton fields as trap crop. Because *L. pratensis* adults

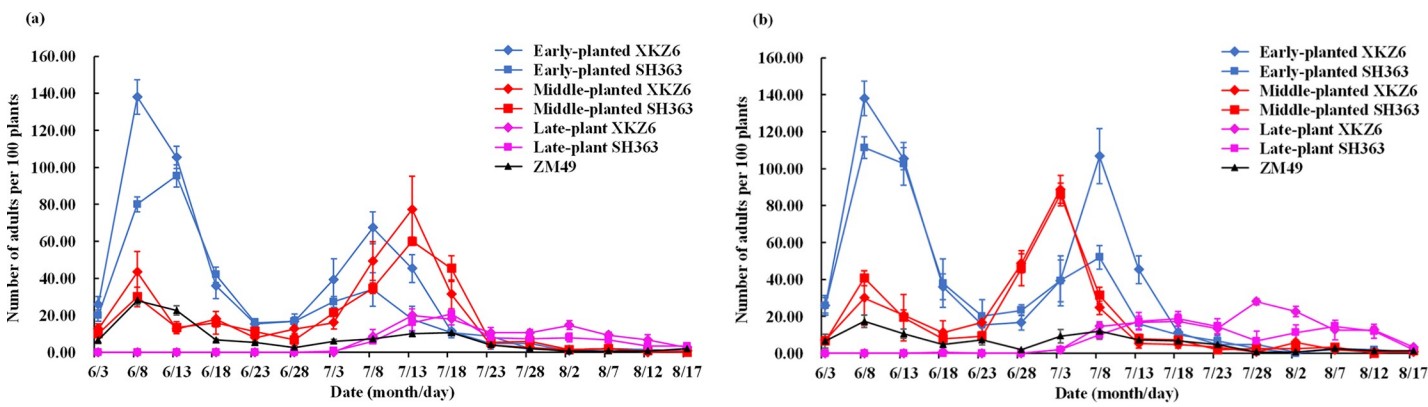

**Fig 2. Population dynamics of *Lygus pratensis* adults on field plots-established with cotton and sunflower with different planting dates in 2015 (a) and 2016 (b).** ZM49 is cotton, XKZ6 and SH363 are sunflower varieties. The same as below.

produced second and three generations in cotton fields, to assess the role of sunflower strips on population dynamics of *L. pratensis* in cotton fields, the abundance including adults and nymphs were investigated.

As compared with control treatments, *L. pratensis* population levels were markedly lower in cotton fields with sunflower borders during 2016 and 2017 (Fig 4). More specifically, significant differences were recorded between fields with and without sunflower strips for population levels of *L. pratensis* nymphs (repeated measures ANOVA; $F_{(1,3)} = 11.44$, $P = 0.0430$) and all individuals (nymphs and adults; $F_{(1,3)} = 23.52$, $P = 0.0167$) in 2016. In 2017, significant differences were recorded for adults ($F_{(1,3)} = 21.36$, $P = 0.0191$), nymphs ($F_{(1,3)} = 10.46$, $P = 0.0481$) and all individuals ($F_{(1,3)} = 34.87$, $P = 0.0097$) in 2017. During both years, population abundances of adults, nymphs and all individuals significantly varied between different sampling periods, and the interactions between trap cropping treatment and sampling period also were significantly for mirid bug abundances (except the adult in 2016; $F_{(8,48)} = 0.63$, $P = 0.7504$) (Table 1).

In both years, cotton fields with sunflower strips had significant reductions in the mean abundance of adults (non-paired t-tests; 2016: t = 6.75, df = 6, $P = 0.0005$; and 2017: t = 4.78, df = 6, $P = 0.0031$), nymphs (2016: t = 7.48, df = 6, $P = 0.0003$; and 2017: t = 4.03, df = 6,

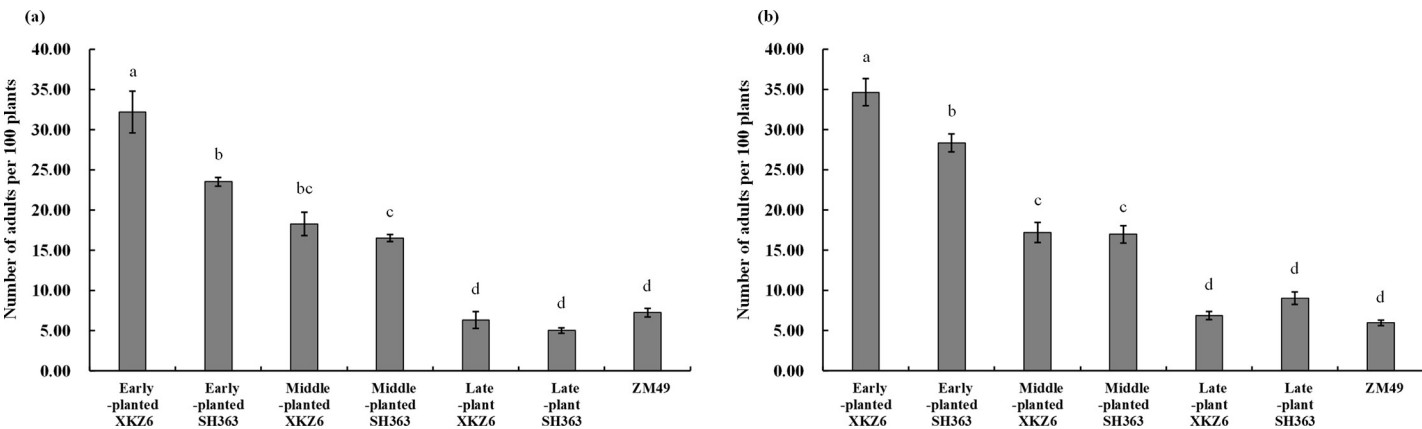

**Fig 3.** Mean population abundances of *Lygus pratensis* adults on field plots-established with cotton and sunflower with different planting dates in 2015 (a) and 2016 (b).

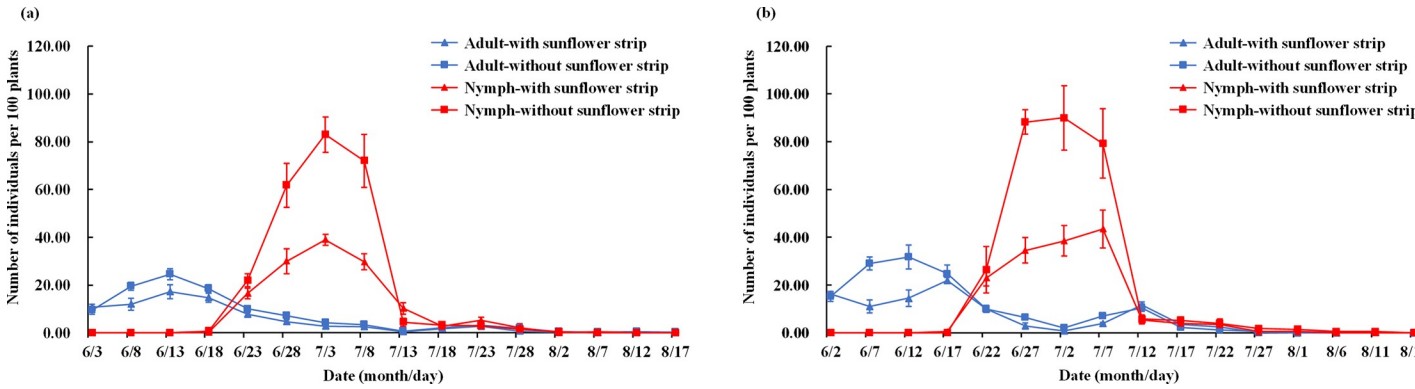

**Fig 4. Population dynamics of *Lygus pratensis* adults and nymphs in cotton fields with and without sunflower strips in 2016 (a) and 2017 (b).** Population dynamics of *Lygus pratensis* adults and nymphs are both in cotton fields which with and without sunflower strip.

*P* = 0.0069), and all individuals (2016: t = 8.78, df = 6, *P* = 0.0001; and 2017: t = 5.83, df = 6, *P* = 0.0011). Also, significant reductions were recorded in the rate of leaf damage (2016: t = 3.35, df = 6, *P* = 0.0154; and 2017: t = 3.93, df = 6, *P* = 0.0077), boll damage (2016: t = 7.26, df = 6, *P* = 0.0003; and 2017: t = 7.24, df = 6, *P* = 0.0004), while the number of bolls was significantly higher (2016: t = 2.94, df = 6, *P* = 0.0259; and 2017: t = 2.47, df = 6, *P* = 0.0484) in the cotton fields with sunflower strips. More specifically, as compared with control treatments, *L. pratensis* abundance decreased by 41.9% and 44.0% in the trap-cropped fields during 2016 and 2017, respectively. Moreover, respective rates of leaf damage and boll damage decreased by 27.3% and 44.8% in 2016, and by 30.6% and 46.0% in 2017. The number of bolls was 8.0% and 7.5% higher in trap-cropped fields during 2016 and 2017, respectively (Table 2).

**Table 1. Mixed effect linear model (MIXED) analysis of population abundance of *Lygus pratensis* in cotton fields with and without sunflower strips, during 2016 and 2017.**

| Year | Development stage | Effect | *Ndf* | *Ddf* | *F* | *P* |
|---|---|---|---|---|---|---|
| 2016 | Adults | Trap cropping treatment | 1 | 3 | 8.44 | 0.0622 |
| | | Sampling period | 8 | 48 | 46.41 | <0.0001 |
| | | Trap cropping treatment * Sampling period | 8 | 48 | 0.63 | 0.7504 |
| | Nymphs | Trap cropping treatment | 1 | 3 | 11.44 | 0.0430 |
| | | Sampling period | 8 | 48 | 339.89 | <0.0001 |
| | | Trap cropping treatment * Sampling period | 8 | 48 | 5.47 | <0.0001 |
| | Adults and nymphs | Trap cropping treatment | 1 | 3 | 23.52 | 0.0167 |
| | | Sampling period | 8 | 48 | 47.80 | <0.0001 |
| | | Trap cropping treatment * Sampling period | 8 | 48 | 4.71 | 0.0003 |
| 2017 | Adults | Trap cropping treatment | 1 | 3 | 21.36 | 0.0191 |
| | | Sampling period | 8 | 48 | 45.39 | <0.0001 |
| | | Trap cropping treatment * Sampling period | 8 | 48 | 2.92 | 0.0098 |
| | Nymphs | Trap cropping treatment | 1 | 3 | 10.46 | 0.0481 |
| | | Sampling period | 8 | 48 | 307.63 | <0.0001 |
| | | Trap cropping treatment * Sampling period | 8 | 48 | 3.93 | 0.0012 |
| | Adults and nymphs | Trap cropping treatment | 1 | 3 | 34.87 | 0.0097 |
| | | Sampling period | 8 | 48 | 25.20 | <0.0001 |
| | | Trap cropping treatment * Sampling period | 8 | 48 | 4.35 | 0.0005 |

**Table 2. Abundance and damage of *Lygus pratensis* to cotton plants in fields with and without sunflower strips, during 2016 and 2017.**

| Year | Abundance and damage | Cotton fields | | Statistic results | | |
|------|---------------------|---------------|---|-------------------|---|---|
| | | With sunflower strip | Without sunflower strip | t | df | P |
| 2016 | Mean number of adults per 100 plants | 8.10±0.11 | 10.85±0.39 | 6.75 | 6 | 0.0005 |
| | Mean number of nymphs per 100 plants | 13.98±0.42 | 27.10±1.70 | 7.48 | 6 | 0.0003 |
| | Mean number of adults and nymphs per 100 plants | 22.08±0.32 | 37.98±1.78 | 8.78 | 6 | 0.0001 |
| | Rate of leaf damage (%) | 9.60±0.43 | 13.20±0.98 | 3.35 | 6 | 0.0154 |
| | Number of bolls per 100 plants | 355.50±5.84 | 329.25±6.75 | 2.94 | 6 | 0.0259 |
| | Rate of boll damage (%) | 22.63±1.81 | 40.99±1.75 | 7.26 | 6 | 0.0003 |
| 2017 | Mean number of adults per 100 plants | 10.38±0.69 | 15.20±0.74 | 4.78 | 6 | 0.0031 |
| | Mean number of nymphs per 100 plants | 16.15±1.09 | 32.20±3.83 | 4.03 | 6 | 0.0069 |
| | Mean number of adults and nymphs per 100 plants | 26.53±1.53 | 47.40±3.24 | 5.83 | 6 | 0.0011 |
| | Rate of leaf damage (%) | 10.45±0.84 | 15.05±0.81 | 3.93 | 6 | 0.0077 |
| | Number of bolls per 100 plants | 348.50±7.31 | 324.25±6.55 | 2.47 | 6 | 0.0484 |
| | Rate of boll damage (%) | 24.60±1.33 | 45.54±2.48 | 7.24 | 6 | 0.0004 |

Statistical comparisons refer to data within a row.

## Discussion

The mirid bug *Lygus pratensis* is a polyphagous, economically-important pest of multiple crops in Europe, central Asia and north Africa, and is a common target of insecticide applications in cotton fields in China. In this study, we revealed how *L. pratensis* adults greatly preferred sunflower plants sown in early season, and showed how sunflower strips clearly suppressed *L. pratensis* population levels and crop damage in cotton. Our work can constitute the basis for further development of integrated pest management schemes against *L. pratensis* in China's cotton crop, and eventually permit significant reductions in insecticide use against this pest.

Aside from its role in lowering *L. pratensis* population levels in China's cotton crop, sunflower has also shown potential as a trap crop in various other crop x pest systems e.g., for control of the stalk borer *Dectes texanus* LeConte (Coleoptera: Cerambycidae) in soybean [33], the brown marmorated stink bug, *Halyomorpha halys* Stål (Hemiptera: Pentatomidae), in pepper [34,35], and the European tarnished plant bug, *L. rugulipennis*, in glasshouse cucumber [30]. Also, multi-species strips of sunflower, Chinese cabbage, marigolds and rapes can help suppress the pollen beetle, *Meligethes aeneus* F. (Coleoptera: Nitidulidae), in cauliflower fields [36]. Hence, considering the importance of *L. pratensis* as a key pest of alfalfa, vegetables, fruit trees, and multiple other crops [1], sunflower strips potentially could also serve as a trap crop and help alleviate insecticide application pressure in these other crops.

The success of trap cropping depends upon the presence of a highly-attractive trap crop, during times when the population of the target pest is high [18]. Hence, the spatio-temporal presence of a trap crop (e.g., planting date, growth dynamics and flowering time) is an important selection criterion when devising a trap-cropping strategy aimed at a particular pest [20]. In certain cases, the target pest can sustain high population levels during prolonged periods of time, and a staggered establishment of a trap crop might thus be needed to sustain its attractiveness [31,37,38]. In our study, the 2015–2016 survey showed comparatively high *L. pratensis* adult infestation levels in cotton fields during June, which were similar to those in plots with early-planted sunflower. The attractiveness of early-planted sunflower also appeared to be superior to that of middle- and late-planted ones, further accentuating its potential as a trap crop for *L. pratensis* in local cotton agro-ecosystems.

As considerable intra-specific variability may occur in the degree of attractiveness of a particular trap crop to a given pest [39,40], field work is often required to carefully select the most attractive varieties. Furthermore, yield and marketability of a given trap crop variety could also be determining factors in securing its wider adoption by farmers [41]. This study compared two common varieties of sunflower and revealed the superior attractiveness of early-planted XKZ6. Yet, no intra-specific differences in in-field *L. pratensis* attraction were reported for middle- and late-planted sunflower, possibly related to particularities of *L. pratensis* adult host plant selection and its marked preference for plants at the budding and blossoming stage [1,16]. Early-planted sunflower generally entered into bud stage by late May, at which time high *L. pratensis* adult abundance was recorded in the field. Also, XKZ6 is characterized by a comparatively slower development from bud to florescence, which might have further enhanced its attractiveness to *L. pratensis* over the course of June.

Periodic suppression of target pests within the trap-crop strips can reduce the potential of re-colonization of the main crop and boost its pest control efficiency [18]. In our study, insecticide sprays in sunflower strips at a 10-day frequency significantly lowered *L. pratensis* abundance in the neighboring cotton field. As sunflower strips solely occupied 6% of the cotton area, the current 6 insecticide sprays directed to the trap-crop strips amounted to a mere 0.36 applications in terms of total cotton area. Thus, trap cropping might greatly reduce the amount of chemical insecticides for *L. pratensis* control in cotton and lower the environmental burden of commercial cotton production in Xinjiang, China.

Sunflower is an important crop in its own right in southern Xinjiang, and socio-economic factors possibly can further facilitate its field-level establishment by local cotton growers. Yet, given the proven environmental impacts of neonicotinoid insecticides such as acetamiprid and imidacloprid [42], and the unmistakable risks those products pose to pollinators, insect natural enemies and other wildlife, future studies should investigate biopesticides and more environmentally-friendly alternatives for *L. pratensis* control. A number of other control options of *L. pratensis*, including the use of sex pheromones and light traps [1], have the potential to be coupled with trap cropping and might further boost control efficacy of sunflower strips. Also, more scientific attention can be paid to the exact size and physical placement of the trap crop, and how such relates to its trapping efficiency and field-level *L. pratensis* pest control [18,43].

During 2016–2017 trials, *L. pratensis* nymphal abundance of subsequent generations was decreased on trap-cropped cotton fields and the associated leaf and boll damage rates of leaf and boll were lower than in control plots. Yet, *L. pratensis* abundance in trap-cropped fields still exceeded existing economic thresholds for this pest (i.e., 12, 20, 41 bugs per 100 plants at budding, blossoming and bolling stages, respectively) in local cotton fields [14]. It is not uncommon for trap cropping to be unable to fully suppress the target pest below economically-significant levels [18,44]. For instance, insecticide application was used in cotton fields with mungbean strips when *A. lucorum* surpassed certain population levels [31]. Hence, trap cropping for *L. pratensis* may need to be complemented with other control tactics, such as targeted applications of selective insecticides [1,15], pheromone-based and light-trap mass-trapping schemes (see above), augmentative biological control, or ecological engineering measures.

Our study validates the usage of sunflower as a trap crop for *L. pratensis* in cotton fields in southern Xinjiang, and describes the particular role of early-planted XKZ6 sunflower strips (coupled with targeted insecticide sprays) to suppress field populations of this mirid bug. Our work provides an effective *L. pratensis* pest management tool which can now be adapted and integrated with other tactics, in order to advance sustainable, environmentally-sound crop protection in China's commercial cotton production.

## Acknowledgments

We extend thanks to Prof. Yanhui Lu (State Key Laboratory for Biology of Plant Diseases and Insect Pests, Institute of Plant Protection, Chinese Academy of Agricultural Sciences, China) for his help and advice in the experimental design and also comments on an early version of this manuscript.

## Author Contributions

**Conceptualization:** Ju Yao.

**Data curation:** Wei Wang.

**Funding acquisition:** Dengyuan Wang, Ju Yao.

**Investigation:** Renfu Zhang.

**Methodology:** Haiyang Liu.

**Project administration:** Dengyuan Wang.

**Software:** Wei Wang, Ju Yao.

**Supervision:** Dengyuan Wang, Ju Yao.

**Writing – original draft:** Renfu Zhang, Dengyuan Wang, Ju Yao.

**Writing – review & editing:** Dengyuan Wang.

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
