## [Decision Letter · Decision Letter 0]

17 Jan 2020

PONE-D-19-30008

Sunflower trap cropping lowers infestation pressure and damage of Lygus pratensis in cotton fields

PLOS ONE

Dear Dr. Yao,

Thank you for submitting your manuscript to PLOS ONE. After careful consideration, we feel that it has merit but does not fully meet PLOS ONE’s publication criteria as it currently stands. Therefore, we invite you to submit a revised version of the manuscript that addresses the points raised during the review process.

I agree with the reviewer's idea. Please improve the manuscript thoroughly according to the reviewer's comments.

We would appreciate receiving your revised manuscript by Mar 02 2020 11:59PM. To enhance the reproducibility of your results, we recommend that if applicable you deposit your laboratory protocols in protocols.io, where a protocol can be assigned its own identifier (DOI) such that it can be cited independently in the future. For instructions see: http://journals.plos.org/plosone/s/submission-guidelines#loc-laboratory-protocols

We look forward to receiving your revised manuscript.

Kind regards,

Yonggen Lou

Academic Editor

PLOS ONE

Journal Requirements:

"This study was supported by National Key R&D Program of China (http://www.most.gov.cn/) (2017YFD0201900), Key R&D Program of Xinjiang Uygur Autonomous Region (http://www.xjkjt.gov.cn/) (2016B01001-6), the National Natural Science Foundation of Xinjiang Uygur Autonomous Region (http://www.xjkjt.gov.cn/) (2019D01A65), the Basic Research Operating Expenses Program of Public Welfare Research Institute from Xinjiang Uygur Autonomous Region (http://www.xjkjt.gov.cn/) (KY2017067). The funders had no role in the study design, data collection and analysis, decision to publish, or preparation of the manuscript."

We note that you have provided funding information that is not currently declared in your Funding Statement. However, funding information should not appear in other areas of your manuscript. We will only publish funding information present in the Funding Statement section of the online submission form.

Reviewers' comments:

Reviewer's Responses to Questions

**Comments to the Author**

1. Is the manuscript technically sound, and do the data support the conclusions?

Reviewer #1: No

2. Has the statistical analysis been performed appropriately and rigorously? 

Reviewer #1: No

3. Have the authors made all data underlying the findings in their manuscript fully available?

Reviewer #1: No

4. Is the manuscript presented in an intelligible fashion and written in standard English?

Reviewer #1: Yes

5. Review Comments to the Author

Reviewer #1: Sunflower trap cropping lowers infestation pressure and damage of Lygus pratensis in cotton fields

The authors addressed a popular topic of agroecology. They found that sunflower strips can be a trapping plant to the mirid bug Lygus pratensis. Also, sunflower strips could significantly decrease abundance of L. pratensis in cotton field. The results are interesting and important. However, the writing is hard to be understood. I suggest the logistic and structure of the manuscript should be substantial improved. Also, I have several concerns below:

You should describe insect sampling methods very carefully. “In each plot, five points were randomly selected and 10 plants were checked per point. On each plant, the number of L. pratensis adults was visually determined and recorded”. So, 50 plants were sampled in each field, right? Maybe a sampling diagram is better to show youe method. Also, you only counted the adults, or bothadult and nymphs . There are also many nymphs in field. You did not show the nymph data in Figure 1. Why you ignore nymphs? You should add more information here.

About the field investigation, you have both sunflower and cottons. Do you investigate the abundance of Lygus pratensis in both crops? In figure 1, you showed the number of adults. Is that cotton? ZM 49 is cotton. I am confusing. I guess you investigate cotton. It makes no sense if you compare the differences between sunflowers and cotton.

You designed three treatments of planting time. In figure 3, do you combine all different time. It is inappropriate that you combine different treatment because late-planted sunflower seems more effective to control Lygus pratensis. I am also confusing that you had nymph data here. You never showed nymph data before.

The discussion is too broad.

Conclusion section. You said that early-planted XKZ6 sunflower strips (coupled with targeted insecticide sprays) to suppress field populations of bug. But I saw the abundance in cotton planted XKZ6 sunflower strips is highest based on Figure 1. Is that right?

6. PLOS authors have the option to publish the peer review history of their article (what does this mean?). If published, this will include your full peer review and any attached files.

Reviewer #1: No

---

## [Author Response · Author response to Decision Letter 0]

26 Mar 2020

I. Responses to the comments and suggestions of Referee

1. You should describe insect sampling methods very carefully. “In each plot, five points were randomly selected and 10 plants were checked per point. On each plant, the number of L. pratensis adults was visually determined and recorded”. So, 50 plants were sampled in each field, right? Maybe a sampling diagram is better to show your method. 

Also, you only counted the adults, or both adult and nymphs. There are also many nymphs in field. You did not show the nymph data in Figure 1. Why you ignore nymphs? You should add more information here.

Reply: In term of the sampling methods, there is a writing error, we have corrected our mistake. Line108-109: changed to “In each plot, using five-point sampling method and 20 consecutive plants were checked each point”. Also, a sampling diagram (figure 1) is generated to show our method clearly.

Beginning in early June, as these first generation of Lygus pratensis nymphs become adults on spring host plants, they disperse to other host plants and then produce second generation there. In other words, before they attack the sunflower and cotton, no L. pratensis’ infestation was occurred on these two crops. Therefore, we only investigate the relative attractiveness of sunflowers and cottons to adults from entering period, ignore the nymphs here. Line110-114.

2. About the field investigation, you have both sunflower and cottons. Do you investigate the abundance of Lygus pratensis in both crops? In figure 1, you showed the number of adults. Is that cotton? ZM 49 is cotton. I am confusing. I guess you investigate cotton. It makes no sense if you compare the differences between sunflowers and cotton.

Reply: In our work, we indeed investigated the abundance of Pratensis in both sunflower and cottons. To clarify our meaning, we have now including the following in Figure 2 legends on Line 459 “Note：ZM49 is cotton, XKZ6 and SH363 are sunflower varieties.”

The aim of this research is to examine the potential of usage of sunflower as a trap crop for L. pratensis in cotton fields. Here, cotton is regarded as control group. Therefore, comparing the abundance of L. pratensis between sunflowers and cotton is requisite and meaningful. On Lines 166-169.

3. You designed three treatments of planting time. In figure 3, do you combine all different time? It is inappropriate that you combine different treatment because late-planted sunflower seems more effective to control Lygus pratensis. I am also confusing that you had nymph data here. You never showed nymph data before.

Reply: In figure 4, we only used the early-planted XKZ6 rather than combing all different time. The aim of the initial field-plot trials is to determine the suitable inter-planted sunflower strips, and the results have indicated the early-planted XKZ6 more effective to attract Lygus pratensis adults, in this context, this treatment was undoubtedly used for subsequent analysis. 

The emphases of these two treatments are different, in figure 2, first generation of L. pratensis as reproductive adults dispersed to other plants, and then produced second generation there. Therefore, we ignore nymphs here. In figure 4, to assess the role of sunflower strips on population dynamics of L. pratensis in cotton fields, the abundance of L. pratensis including adults and nymphs were investigated. 

4. The discussion is too broad. Conclusion section. You said that early-planted XKZ6 sunflower strips (coupled with targeted insecticide sprays) to suppress field populations of bug. But I saw the abundance in cotton planted XKZ6 sunflower strips is highest based on Figure 1. Is that right?

Reply: The objective of our work is to assess the relative attractiveness of sunflower (Helianthus annuus) to Lygus pratensis adults in local cotton plots. As shown in Figure 1, the abundance in cotton early-planted XKZ6 sunflower plots is highest, thus we concluded that establishment of sunflower strips could contribute to suppress field populations of bug in cotton crops. Our conclusion is coincident with figure 2.

The conclusion section was outlined in Discussion (L203-L209): “Our study validates the usage of sunflower as a trap crop for L. pratensis in cotton fields in southern Xinjiang, and describes the particular role of early-planted XKZ6 sunflower strips (coupled with targeted insecticide sprays) to suppress field populations of this mirid bug.”

---

## [Decision Letter · Decision Letter 1]

23 Apr 2020

PONE-D-19-30008R1

Sunflower trap cropping lowers infestation pressure and damage of Lygus pratensis in cotton fields

PLOS ONE

Dear Yao,

Thank you for submitting your manuscript to PLOS ONE. After careful consideration, we feel that it has merit but does not fully meet PLOS ONE’s publication criteria as it currently stands. Therefore, we invite you to submit a revised version of the manuscript that addresses the points raised during the review process.

Please improve the manuscript according to the two reviewers' comments.

We would appreciate receiving your revised manuscript by Jun 07 2020 11:59PM. To enhance the reproducibility of your results, we recommend that if applicable you deposit your laboratory protocols in protocols.io, where a protocol can be assigned its own identifier (DOI) such that it can be cited independently in the future. For instructions see: http://journals.plos.org/plosone/s/submission-guidelines#loc-laboratory-protocols

We look forward to receiving your revised manuscript.

Kind regards,

Yonggen Lou

Academic Editor

PLOS ONE

Reviewers' comments:

Reviewer's Responses to Questions

**Comments to the Author**

1. If the authors have adequately addressed your comments raised in a previous round of review and you feel that this manuscript is now acceptable for publication, you may indicate that here to bypass the “Comments to the Author” section, enter your conflict of interest statement in the “Confidential to Editor” section, and submit your "Accept" recommendation.

Reviewer #2: All comments have been addressed

Reviewer #3: (No Response)

2. Is the manuscript technically sound, and do the data support the conclusions?

Reviewer #2: Partly

Reviewer #3: Partly

3. Has the statistical analysis been performed appropriately and rigorously? 

Reviewer #2: I Don't Know

Reviewer #3: Yes

4. Have the authors made all data underlying the findings in their manuscript fully available?

Reviewer #2: Yes

Reviewer #3: No

5. Is the manuscript presented in an intelligible fashion and written in standard English?

Reviewer #2: Yes

Reviewer #3: Yes

6. Review Comments to the Author

Reviewer #2: 1. The title is not good enough. The following two titles are suggested: Field evaluation of sunflower as a potential trap crop of Lygus pratensis in cotton fields or Intercropping sunflower reduces the infestation and damage of Lygus pratensis in cotton fields

2. The introduction is not strong enough in the present version, and more latest literatures (published in the international journals) of intercropping effect on herbivorous pests and the underline mechanisms should be introduced.

3. The diagram in the current version is not the really diagram, and I made no sense from it. Please referred to other published papers.

4. LSD was abandoned in the present. Turkey’s test is suggested to analyze your data.

5. The language of the paper should be polished.

Reviewer #3: In this paper, the occurrences of Lygus pratensis were analyzed between different sunflower and cotton crops in field. Next another test were designed to estimate the influence of L. pratensis population on cotton crops with or without sunflower border strips. It reported that adult abundance of L. pratensis was highest on early-planted sunflower in field, and sunflower could reduce the damage of cotton by L. pratensis. The results is very interesting and important for IPM on cotton cultivation, but the writing and some experiments of this paper need to be further improved. Some comments as below,

1. Why did you use five-point sampling method? As Yao et al (2016) reported that the spatial distribution pattern L. pratensis were aggregated distribution.

2. Please give location of the sample site when using five-point sampling？Especially when collecting data from trap-cropping field with two 1-m wide strips of sunflower on the opposite borders of the field, parallel to the cotton rows.

3. As results of figure 2, the size of plant is important factor for L. pratensis to select plants. It is not very available to say the sunflower more attractive to cotton. Could you please show some other data such as comparing the attractive ability between same size cotton or sunflower.

4. Please check the data of Table 1 that in 2016, the p value of the adults, Nymphs were 0.0622 and 0.0430 respectively, but the p value of the total the adult and nymphs was 0.0167 not in range 0.0430~0.0622, is right?

7. PLOS authors have the option to publish the peer review history of their article (what does this mean?). If published, this will include your full peer review and any attached files.

Reviewer #2: No

Reviewer #3: No

---

## [Author Response · Author response to Decision Letter 1]

3 Jun 2020

Dear editors,

Thank you very much for your attention and careful consideration to our manuscript. We appreciate reviewers very much for their constructive comments and suggestions on our manuscript.

We have carefully considered your and referee’s suggestion and comments and revised our manuscript according to these precious comments. Those comments are invaluable for improving our paper. We have accordingly revised the manuscript carefully and hope that this new version could be accepted for publication in PLOS ONE The main corrections and the responds to the comments point to point are given below:

I. Responses to the comments and suggestions of Referee

1. The title is not good enough. The following two titles are suggested: Field evaluation of sunflower as a potential trap crop of Lygus pratensis in cotton fields or Intercropping sunflower reduces the infestation and damage of Lygus pratensis in cotton fields.

Reply: Thanks for the reviewer's suggestion, and the title has been changed to "Field evaluation of sunflower as a potential trap crop of Lygus pratensis in cotton fields"(Line 5-6)

2. The introduction is not strong enough in the present version, and more latest literatures (published in the international journals) of intercropping effect on herbivorous pests and the underline mechanisms should be introduced.

Reply: Thanks for the suggestion, and it has been supplemented in the revised manuscript.

3. The diagram in the current version is not the really diagram, and I made no sense from it. Please referred to other published papers.

Reply: Thanks for the suggestion, and referred to the other published papers, we have been re-made the diagram in the revised manuscript.

4. LSD was abandoned in the present. Turkey’stest is suggested to analyze your data.

Reply: Thanks for the suggestion. Because the population dynamics of Lygus pratensis on cotton and sunflower are all showed seasonal changes, so the data analysis are as follows: Differences in the abundance of L. pratensis (including adults, nymphs, and total population, as three counts) in cotton and sunflower plots (2015-2016) or in cotton fields with/without sunflower strips (2016-2017) were analyzed by a repeated-measures analysis of variance (ANOVA) using SAS PROC MIXED, and the means were compared by the least significant difference test (LSD). The data were log10(n+1)-transformed before analysis. The mean abundances of L. pratensis adults in plots of cotton and sunflower during 2015-2016 were compared using one-way ANOVA followed by Tukey’s HSD for multiple comparisons. Mean abundances of L. pratensis population, the damage rate of leaves and bolls (arcsin-transformed), and the number of mature bolls in cotton fields with/without sunflower strips during 2016-2017 were compared through non-paired t-tests.

5. The language of the paper should be polished.

Reply: Thanks for the reviewer's suggestion, the language has been improved and polished in the revised manuscript.

II. Responses to the comments and suggestions of Referee 2

1. Why did you use five-point sampling method? As Yao et al (2016) reported that the spatial distribution pattern L. pratensis were aggregated distribution.

Reply: As Jiao et al (2012) reported that the spatial distribution pattern of Apolygus lucorum (Hemiptera: Miridae) is aggregated distribution. five-point sampling method is usually used for the investigation and sampling of Apolygus lucorum in fields (Wu et al, 2002; Lu et al, 2008; Lu et al, 2009), so we borrowed their method, using five-point sampling method in this paper.

2. Please give location of the sample site when using five-point sampling？Especially when collecting data from trap-cropping field with two 1-m wide strips of sunflower on the opposite borders of the field, parallel to the cotton rows.

Reply: In order to show our sampling method more intuitively, we made a sampling diagram (Figure 1b)

3. As results of figure 2, the size of plant is important factor for L. pratensis to select plants. It is not very available to say the sunflower more attractive to cotton. Could you please show some other data such as comparing the attractive ability between same size cotton or sunflower.

Reply: As you said, the size of plant is important factor for L. pratensis to select plants. In this paper, the two most common sunflower varieties (XKZ6 and SH363) were selected to investigated the abundance of Lygus pratensis with different varieties and planting dates, and cotton was investigated at the same time. According to previous habit, sunflower and cotton were all investigated 100 plants each time. But ignore the impact of plant size, this is indeed the negligence of our work.

And on the basis, we investigated the population of Lygus pratensis on sunflower(XKZ6) and cotton with the same size in 2017. The result showed that, compared to cotton, sunflowers have more attractive ability to Lygus pratensis. But this data has already appeared in another submitted paper.

4. Please check the data of Table 1 that in 2016, the p value of the adults, Nymphs were 0.0622 and 0.0430 respectively, but the p value of the total the adult and nymphs was 0.0167 not in range 0.0430~0.0622, is right?

Reply: In response to your questions, we checked and analyzed the data again.

Compared with control treatments, population levels of L. pratensis adults and nymphs were markedly lower in cotton fields with sunflower borders. So the population levels of L. pratensis all individuals (nymphs and adults) were lower even more in cotton fields with sunflower borders, compared with control treatment. Therefore, the p value of the total adults and nymphs was 0.0167 not in range 0.0430~0.0622.

We hope that the revised manuscript will meet with formal acceptance. 

Again, we thank you and the reviewers for your time and effort in reviewing this manuscript.

Kindest regards.

Yours,

Renfu Zhang 

Institute of Plant Protection

Xinjiang Academy of Agricultural Sciences

No.403, Nanchang Road

Urumqi, Xinjiang 830091, China

---

## [Decision Letter · Decision Letter 2]

22 Jun 2020

PONE-D-19-30008R2

Field evaluation of sunflower as a potential trap crop of Lygus pratensis in cotton fields

PLOS ONE

Dear Dr. Yao,

Thank you for submitting your manuscript to PLOS ONE. After careful consideration, we feel that it has merit but does not fully meet PLOS ONE’s publication criteria as it currently stands. Therefore, we invite you to submit a revised version of the manuscript that addresses the points raised during the review process.

Please improve the manuscript according to the comments from the reviewer 3.

We look forward to receiving your revised manuscript.

Kind regards,

Yonggen Lou

Academic Editor

PLOS ONE

Reviewers' comments:

Reviewer's Responses to Questions

**Comments to the Author**

1. If the authors have adequately addressed your comments raised in a previous round of review and you feel that this manuscript is now acceptable for publication, you may indicate that here to bypass the “Comments to the Author” section, enter your conflict of interest statement in the “Confidential to Editor” section, and submit your "Accept" recommendation.

Reviewer #2: All comments have been addressed

Reviewer #3: (No Response)

2. Is the manuscript technically sound, and do the data support the conclusions?

Reviewer #2: Yes

Reviewer #3: Yes

3. Has the statistical analysis been performed appropriately and rigorously? 

Reviewer #2: Yes

Reviewer #3: Yes

4. Have the authors made all data underlying the findings in their manuscript fully available?

Reviewer #2: Yes

Reviewer #3: Yes

5. Is the manuscript presented in an intelligible fashion and written in standard English?

Reviewer #2: Yes

Reviewer #3: Yes

6. Review Comments to the Author

Reviewer #2: All questions has been answered and all responses meet formatting specifications. Thank you for your revison.

Reviewer #3: It is interesting results of this text that sunflower can be an effective trap crop for L. pratensis in field. The writing of the paper should to be polished and concise before be published, and other some suggest or question:

1.Line 35, integrated pest management (IPM), please capitalize the first letter of the first word.

2. in text, there are some presentation like "F=70.88，df=6,14， P<0.0001" ,you can try like"F(6,14)=70.88, P=... "

3.Line 160, "F=70.88，df=6,14， P<0.0001", the sampling times is 16 in range 6/3 to 8/17 in Figure 2, please check "df=14" is right?

4. Also please check Ndf number of sampling period=8, you need to explain how to select data for analysis. please see times of sampling is 11 as showed in figure 4.

7. PLOS authors have the option to publish the peer review history of their article (what does this mean?). If published, this will include your full peer review and any attached files.

Reviewer #2: No

Reviewer #3: No

---

## [Author Response · Author response to Decision Letter 2]

23 Jul 2020

1. Line 35, integrated pest management (IPM), please capitalize the first letter of the first word.

Reply: Thanks for the referee's suggestion. We have capitalized the first letter of the first word of integrated pest management (IPM) (Line 35).

2. in text, there are some presentation like "F=70.88, df=6,14, P<0.0001", you can try like" F (6,14) =70.88, P=... "

Reply: Thanks for the referee's suggestion. We have modified the format like your suggestion (Line 156-160 and 175-182).

3.Line 160, "F=70.88，df=6,14, P<0.0001", the sampling times is 16 in range 6/3 to 8/17 in Figure 2, please check "df=14" is right?

Reply: Thanks for the referee's suggestion. Population levels of L. pratensis adults among all seven treatments were analyzed using repeated measures ANOVA in 2015 and 2016, each of the above treatments was established in 3 field plots (i.e., replicates). it means that, 7 treatments, each treatment replicates 3 times, total of 21 plots in this test. Therefore, the correct is df=6,12. Through analysis again, found that this indeed a mistake. Thank you very much for your attention and careful review(Line 175-182).

4. Also please check Ndf number of sampling period=8, you need to explain how to select data for analysis. please see times of sampling is 11 as showed in figure4.

Reply: Thanks for the referee's suggestion. Difference in the abundance of L. pratensis (including adults, nymphs, and total population, as three counts) in cotton fields with/without sunflower strips (2016-2017) were analyzed by a repeated-measures analysis of variance (ANOVA) using SAS PROC MIXED, and the means were compared by the least significant difference test (LSD). In 2016 and 2017, L. pratensis abundances in both cotton fields were below the existing economic thresholds for this pest (i.e., 20 bugs per 100 cotton plants at blossoming stage; Wang et al., 2016) after middle July. Hence, only population abundance records from early June to mid-July were included in the repeated-measures ANOVA and non-paired t-tests, select sampling time 8 times for statistical analysis. Detailed explanation showed in the statistical analysis (line 134-147).

---

## [Editor Report · Decision Letter 3]

27 Jul 2020

Field evaluation of sunflower as a potential trap crop of Lygus pratensis in cotton fields

PONE-D-19-30008R3

Dear Dr. Yao,

We’re pleased to inform you that your manuscript has been judged scientifically suitable for publication and will be formally accepted for publication once it meets all outstanding technical requirements.

Kind regards,

Yonggen Lou

Academic Editor

PLOS ONE
---

## [Editor Report · Acceptance letter]

5 Aug 2020

PONE-D-19-30008R3 

Field evaluation of sunflower as a potential trap crop of *Lygus pratensis* in cotton fields 

Dear Dr. Yao:

I'm pleased to inform you that your manuscript has been deemed suitable for publication in PLOS ONE. Congratulations! Your manuscript is now with our production department. 

Kind regards, 

on behalf of

Dr. Yonggen Lou 

Academic Editor

PLOS ONE